# Small peptide CSF fingerprint of amyotrophic lateral sclerosis

Rea Lumi[1], Susanne Petri[1], Justyna Siwy[2], Agnieszka Latosinska[2], Julia Raad[2], Petra Zürbig[2], Thomas Skripuletz[1], Harald Mischak[2], Joachim Beige[3,4,5]*

1 Department of Neurology, Hannover University Medical School, Hannover, Germany, 2 Mosaiques Diagnostics GmbH, Hannover, Germany, 3 Kuratorium for Dialysis and Kidney Transplantation, Neu-Isenburg, Germany, 4 Martin-Luther-University Halle/Wittenberg, Halle/Saale, Germany, 5 Hospital Sankt Georg gGmbH, Leipzig, Germany

☯ These authors contributed equally to this work.
* Joachim.Beige@kfh.de

## Abstract

### Background

Amyotrophic lateral sclerosis (ALS) is a neurodegenerative disease characterized by abnormal protein aggregation in the motor neurons. Present and earlier proteomic studies to characterize peptides in cerebrospinal fluid (CSF) associated with motoneuron pathology did not target low molecular weight proteins and peptides. We hypothesized that specific changes in CSF peptides or low molecular weight proteins are significantly altered in ALS, and that these changes may support deciphering molecular pathophysiology and even guide approaches towards therapeutic interventions.

### Methods

Cerebrospinal fluid (CSF) from 50 ALS patients and 50 non-ALS controls was collected, centrifuged immediately after collection, aliquoted into polypropylene test tubes, frozen within 30–40 min after the puncture, and stored at −80˚C until use. Peptides were sequenced using capillary electrophoresis or liquid chromatography/mass spectrometry (CE-MS/MS or LC-MS/MS).

### Findings

In the CSF of 50 patients and 50 non-ALS controls 33 peptides were found, of which 14 could be sequenced using a non-lytic single-pot proteomic detection method, CE/MS. ALS deregulated peptides vs. controls included Integral membrane protein 2B, Neurosecretory protein VGF, Osteopontin, Neuroendocrine protein 7B2 (Secretogranin-V), EGF-containing fibulin-like extracellular matrix protein 1, Xylosyltransferase 1 XT-1, Chromogranin-A, Superoxide dismutase SOD-1, Secretogranin-1 (Chromogranin B), NR2F2 Nuclear Receptor Subfamily 2 Group F Member 2 and Collagen alpha-1(VII) chain.

**Data Availability Statement:** Data (peptide sequences) are included in the manuscript. All detected peptides were deposited, matched, and annotated in a Microsoft SQL database allowing

further statistical analysis (https://zenodo.org/records/10716167).

**Funding:** The author(s) received no specific funding for this work.

**Competing interests:** S. Petri received honoraria as speaker/consultant from Biogen GmbH, Roche, Novartis, Teva, Cytokinetics Inc., Desitin, Italfarmaco, Ferrer, Amylyx, and Zambon; and grants from DGM e.V, Federal Ministry of Education and Research, German Israeli Foundation for Scientific Research and Development, EU Joint Program for Neurodegenerative Disease Research. J. Beige received funding from GSK and German Federal Ministries of Research and Health. This does not alter our adherence to PLOS ONE policies on sharing data and materials.

## Interpretation

Most striking deregulations in CSF from ALS patients were found in VGF, Osteopontin, SOD-1 and EFEMP1 peptides. No associations of disease severity, duration and region of onset with sequenced peptides were found.

## Introduction

Amyotrophic lateral sclerosis (ALS) is a neurodegenerative disease characterized by the progressive loss of upper and lower motor neurons in the primary motor cortex, brain stem and spinal cord leading to rapidly progressive atrophy and paralysis of all skeletal muscles including the diaphragm and ultimately resulting in death by respiratory failure within few years from symptom onset.

About 90% of ALS cases occur sporadically while 5–10% are familial. Despite an evolving understanding of the disease pathophysiology, effective disease modifying treatments for ALS remain elusive, and much research is ongoing to find ways to slow or halt the progression of the disease [1].

Abnormal protein aggregates in motor neurons are the neuropathological hallmark of ALS. The DNA- and RNA-binding protein TDP-43 is a major component of these protein aggregates and can be detected in the cytoplasm of motor neurons in sporadic and most of the familial ALS cases. Numerous genes have been linked to familial ALS, with the most common being mutations in the C9ORF72, SOD1, TARDBP, and FUS genes [2]. The SOD1 gene codes for an enzyme that neutralizes free radicals, and mutations in this gene can reduce the enzyme's ability to ameliorate oxidative stress and increase protein aggregation.

The pathophysiology of ALS is multifaceted and complex and various interacting mechanisms such as disruption of axonal transport processes, mitochondrial dysfunction, endoplasmic reticulum (ER) stress, excitotoxicity, disturbed RNA homeostasis and activation of glial cells, such as astrocytes and microglia, contribute to motor neuron degeneration [1,3,4].

Whether proteome distribution patterns in cerebrospinal fluid (CSF) are signatory for ALS and may even hold molecular information useful for therapy has been investigated in a multitude of studies [5]. However, these studies did not investigate the low molecular weight proteins and peptides comprehensively. Peptides were found to be highly significantly affected in other neurodegenerative diseases like frontotemporal dementia or Morbus Alzheimer [6,7]. Based on these previous study results, we generated the hypothesis that specific changes in CSF peptides or low molecular weight proteins are significantly changed in ALS, and that these changes may support deciphering molecular pathophysiology and even guide approaches towards therapeutic interventions. We investigated this hypothesis, using capillary electrophoresis coupled to mass spectrometry (CE-MS) after a desalting preparation process, without digestion of the proteins, yielding a largely unchanged CSF sample. CE-MS of these non-modified samples inherits the opportunity to measure a holistic peptidomic pattern up to a peptide/protein mass to about 20 kDa.

## Methods

Patients were recruited between September 2020 and May 2023 at the Department of Neurology, Hannover Medical School, Germany. The study was approved by the Ethics Committee of Hannover Medical School. The vote was initially issued in 2012 to Susanne Petri for the

project "Deutsches Netzwerk für Motoneuronerkrankungen–German Network for Motor Neuron Diseases (MND-Net)", comprising collection and analysis of ALS patient and control data and biomaterials for an unlimited duration. The latest amendment was made in 2018 again for an unlimited period of collection. Written informed consent was obtained from all participants in accordance with the Declaration of Helsinki.

Cerebrospinal fluid (CSF) from 50 ALS patients diagnosed with possible, probable or definite ALS according to the El Escorial criteria [8] (and all of them fulfilling the recently established Gold Coast criteria, [9]) was obtained as part of the differential diagnostics after written informed consent by lumbar puncture in a sitting position according to standard procedures [10,11]. Patient data collection was performed from 06/01 to 10/15/2023. SP and RL had access to information that could identify individual participants during or after data collection.

Disease severity was assessed by the ALS functional rating scale-revised (ALSFRS-r). ALS progression rate (ALS-PR) i.e. monthly decline of ALSFRS-r was assessed as follows: 48 – (ALSFRS-r at initial visit) /disease duration (months).

Patients without neurodegenerative diseases who underwent lumbar puncture for differential diagnosis of facial palsy and headaches resulting in normal parameters of routine CSF analysis (except one control patient with CSF pleocytosis due to neuroborreliosis) served as controls.

## CSF sample collection

After collection of 4 mL CSF for routine diagnosis, additional 5 ml of the CSF was sampled into a polypropylene test tube. The CSF was centrifuged immediately after collection (1600 g, 4˚C, 15 min), aliquoted into polypropylene test tubes (each aliquot, 750 µL), frozen within 30–40 min after the puncture and stored at −80˚C until use. Since blood contaminations may affect the proteome of CSF [6] we controlled our CSF samples for traces of hemoglobin and the presence of erythrocytes by microscopy of a centrifuged CSF sample. Using these two methods we can exclude any blood contamination of our CSF samples above 0·001% by that controlling a potential serious confounder of CSF biomarker studies with regard to proteins abundant in plasma. The CSF was at no time thawed/refrozen.

## CSF sample preparation CE-MS analysis

After thawing, 700 µL of CSF were diluted with 700 µL buffer (pH 10·5; 2 M urea, 100 mmol/L NaCl, 0·0125% NH3; Sigma-Aldrich, Taufkirchen, Germany) and centrifuged using Centrisart ultrafiltration devices (cut off 20 kDa, Sartorius, Göttingen, Germany) at 4˚C until 1·1 mL of filtrate was obtained. Subsequently the filtrate was applied onto a PD-10 desalting column (Amersham Bioscience, Uppsala, Sweden) equilibrated in 0·01% NH4OH in HPLC-grade in water (Roth, Germany) to decrease matrix effects by removing urea, electrolytes, salts, and to enrich peptides present. Finally, the eluate was lyophilized stored at 4˚C and resuspended in 10 µL HPLC-grade water before CE-MS analysis.

CE-MS analyses were accomplished using the P/ACE MDQ capillary electrophoresis system (Beckman Coulter, Fullerton, USA) online coupled to a MicroTOF MS (BrukerDaltonic, Bremen, Germany). The electro-ionization sprayer (Agilent Technologies) was grounded, and the ion spray interface potential was defined between −4 and −4·5 kV. Spectra were accumulated every 3s along with over a range of m/z to 350–3000. Detailed information on accuracy, precision, selectivity, sensitivity, reproducibility and stability of the CE-MS method have been described previously [12].

## CE-MS data processing

After the CE-MS analysis, mass spectral ion peaks representing identical molecules at different charge states were deconvoluted into single masses using a proprietary software. Only signals with $z>1$ observed in a minimum of 3 consecutive spectra with a signal-to-noise ratio of at least 4 were considered. The resulting peak list characterises each peptide by its mass and migration time. Data were calibrated utilising internal standards as reference data points for mass and migration time by applying global and local linear regression, respectively [13]. Reference signals of abundant peptides were used as internal standards for calibration of signal intensity using linear regression. This procedure is highly reproducible and addresses both analytical and dilution variances in a single calibration step. The performance of the analytical platform was assessed and described in detail previously. In short, among 60 independent analytic runs of a single sample, the coefficient of variation was 1%. The obtained peak list characterises each polypeptide by its calibrated molecular mass [Da], calibrated CE migration time [min] and normalised signal intensity. All detected peptides were deposited, matched, and annotated in a Microsoft SQL database allowing further statistical analysis (https://zenodo.org/records/10716167).

## Biomarker identification and statistical analysis

For the identification of potential ALS-related peptide biomarkers, a comparison between ALS cases and controls and regression analyses of peptide abundances with ALS severity were performed. Only peptides that were detected in at least 30% (frequency threshold) of the samples in at least one of the two groups were further considered for statistical analysis. Using the Wilcoxon rank-sum test followed by adjustment for multiple testing with the false-discovery rate method presented by Benjamini and Hochberg [14], adjusted P-values were calculated based on the comparison between cases and healthy controls. Only peptides with a P-value less than 0·05 were considered as statistically significant.

Peptide abundance group comparisons of cases vs. controls, using binomial logistic regression of dichotomized cases by severity (ALSFRSr below vs. above 40), linear regression analyses with neurofilament levels and disease duration as dependent variables were performed with age and sex as covariates (statistical software R).

## Sequencing of peptides

Peptides were sequenced using CE-MS/MS or LC-MS/MS analysis, as described in detail. MS/MS experiments were using an Ultimate 3000 nano-flow system (Dionex/LC Packings, USA) or a P/ACE MDQ capillary electrophoresis system (Beckman Coulter, Fullerton, CA), both connected a Q Exactive™ Plus Hybrid Quadrupole-Orbitrap™ Mass Spectrometer (Thermo-Fisher Scientific, Waltham, Massachusetts, USA). The mass spectrometer is operated in data-dependent mode to automatically switch between MS and MS/MS acquisition. Survey full-scan MS spectra (from m/z 300–2000) were acquired in the Orbitrap. Ions were sequentially isolated for fragmentation. Data files were searched against the UniProt human nonredundant database using Proteome Discoverer 2·4 and the SEQUEST search engine. Relevant settings were: no fixed modifications, oxidation of methionine and proline as variable modifications. FDR was set to 1%, precursor mass tolerance was 5 ppm and fragment mass tolerance 0·05 Da. For further validation of obtained peptide sequences, the correlation between peptide charge at the working pH of 2 and CE-migration time was utilised to minimise false-positive derivation rates [15]. Calculated CE-migration time of the sequence candidate based on its peptide sequence (number of basic amino acids) was compared to the experimental migration time.

## Results

Mean age of the patient cohort was 67·5 ± 10·4 years, 36% with bulbar and 64% with spinal onset (18% cervical, 6% thoracic and 40% lumbosacral). 48% were female. Disease duration prior to CSF collection ranged from 3 months to 72 months, with a mean of 15 months, disease severity rate (ALSFRSr) was 38·9 ± 5·9 and disease progression rate (ALS-PR), Table 1.

In 8% of the patients, mutations in the FUS, C9ORF72 and SOD1 genes have been identified while genetic testing in further 16% of the cases was negative. For the remaining 76% of patients, no genetic information was available, out of these, two had a positive family history for ALS. Controls were 47.0 ± 19.0 yrs and 62% female.

After compiling the peptidomic data of patients vs. controls and after adjustment for multiple testing, 33 peptides were found significantly associated with ALS, with a fold-change between 0.1 and 13.16 compared to controls (Table 2).

In two known familial cases with SOD1 mutations, no quantitative variations of peptidomic patterns compared to the overall ALS dataset was found.

Using mass spectrometric sequence analysis (tandem mass spectrometry, MS/MS) of the discriminating biomarkers, among 33 peptides, 14 could be identified by amino acid sequences (Table 2).

These associations were confirmed in multivariate, both linear and logistic regression of the disease severity scales ALSFRS-r and ALS-PR and neurofilament concentrations in CSF and serum against peptide abundances.

No associations of sequenced peptide abundances with disease severity or progression were found in univariate (Fig 1) and multivariate analyses. Likewise, using light and heavy neurofilament chain serum concentrationss, region of onset and disease severity as dependent variables, no associations were found between peptide markers and disease surrogates. Among non-sequenced peptides, no fragments of neurofilaments could be detected, presumably following the detection threshold of 20 kDa of CE/MS and the larger neurofilament size of 65 kDA.

Beside typical neuropeptides, like Neurosecretory protein VGF, Integral membrane protein 2B, Neuroendocrine protein 7B2, also peptides of Osteopontin, Secretogranin V, Chromogranin A and B, Xylosyltransferase 1, Superoxide dismutase 1, NR2F2 Nuclear Receptor Subfamily 2 Group F Member 2, and Collagen alpha-1(VII) chain were identified.

## Discussion

The objective of this open rapid publication is stimulation of further research regarding the peptides and parental proteins which we found to be differentially regulated in a population of 50 ALS patients compared to peptides in non-ALS individuals.

**Table 1. Anthropometric characteristics, CSF phosphorylated neurofilament heavy chain (pNfH) and serum light chain (s-NfL) concentrations, both pg/mL), disease severity (ALSFRS-r) and disease progression rate (ALS-PR) clustered by neurological region of ALS onset.**

| | Onset | | | | | | | | | | | | | | |
| | bulbar (n = 18) | | | cervical (n = 9) | | | lumbos. (n = 20) | | | thoracic (n = 3) | | | In total (n = 50) | | |
|---|---|---|---|---|---|---|---|---|---|---|---|---|---|---|---|
| Age | 72.2 | ± | 9.6 | 62.6 | ± | 10.8 | 65.3 | ± | 9.9 | 69.0 | ± | 11.3 | 67.5 | ± | 10.4 |
| Male (n, %) | 6 | , | 33 | 6 | , | 67 | 12 | , | 60 | 2 | , | 67 | 26 | , | 52 |
| pNfH | 3011 | ± | 2882 | 390 | ± | 2687 | 3089 | ± | 3607 | 2584.7 | ± | 1369 | 3180.0 | ± | 17.4 |
| s-NfL | 179 | ± | 141 | 152 | ± | 98.9 | 133 | ± | 114 | 140 | ± | 41.9 | 153 | ± | 119 |
| Duration | 27.8 | ± | 13.2 | 33.6 | ± | 13.2 | 43.2 | ± | 20.8 | 30.3 | ± | 0.6 | 35.2 | ± | 30.4 |
| ALS-PR | 1.2 | ± | 1.4 | 0.7 | ± | 0.4 | 0.8 | ± | 0.7 | 0.7 | ± | 0.1 | 0.9 | ± | 0.97 |
| ALSFRS-r | 38.4 | ± | 6.40 | 41.6 | ± | 4.48 | 37.9 | ± | 6.06 | 40.0 | ± | 4.36 | 38.9 | ± | 5.86 |

**Table 2. Marker IDs, molecular mass, amino acids sequence, parental protein, mean abundance (expression) in cases and controls and mean fold change of ALS patients vs. controls (columns in sequence from left to right) in differentially regulated peptides in ALS CSF.**

| Mark | Mass (Da) | Sequence | Parental Protein | Start AA | Stop AA | mean cases | mean ctr | fold change | P |
|---|---|---|---|---|---|---|---|---|---|
| 213 | | | | | | 4·25 | 42·41 | 0·10 | 0·023 |
| 1267 | | | | | | 3·52 | 12·86 | 0·27 | 0·035 |
| 3250 | 1083·501 | FENKFAVET | Integral membrane protein 2B | | | 102·40 | 68·76 | 1·49 | 0·043 |
| 15005 | 1222·63 | AVPGPKDGSAPEV | Neurosecretory protein VGF | 47 | 59 | 14·35 | 67·76 | 0·21 | 0·026 |
| 16933 | 1247·57 | ATDEDITSHME | Osteopontin | 184 | 194 | 22·92 | 86·32 | 0·27 | 0·0014 |
| 22376 | | | | | | 14·91 | 161·33 | 0·09 | 0·0001 |
| 35879 | 1388·74 | SVNPYLQGQRLD | Neuroendocrine protein 7B2 (Secretogranin-V) | 182 | 193 | 578·89 | 144·38 | 4·01 | 0·026 |
| 39327 | | | | | | 53·33 | 12·28 | 4·34 | 0·023 |
| 47940 | 1528·76 | NPADPQRIPSNPSH | EGF-containing fibulin-like extracellular matrix protein 1 | 142 | 155 | 18·68 | 3·63 | 5·15 | 0·042 |
| 52825 | 1613·88 | VGGGEQPPPAPAPRRE | Xylosyltransferase 1 | 51 | 66 | 14·56 | 30·64 | 0·48 | 0·036 |
| 61368 | 1800·02 | APPEPVPPPRAAPAPTHV | Neurosecretory protein VGF | 487 | 504 | 1103·39 | 590·56 | 1·87 | 0·023 |
| 75685 | 2128·97 | EGQEEEEDNRDSSMKLSF | Chromogranin-A | 359 | 376 | 196·33 | 364·91 | 0·54 | 0·049 |
| 77655 | | | | | | 231·93 | 122·08 | 1·90 | 0·027 |
| 87061 | 2430·22 | KANDESNEHSDVIDSQELSKVS | Osteopontin | 249 | 270 | 502·29 | 87·07 | 5·77 | 0·049 |
| 92192 | | | | | | 206·89 | 480·93 | 0·43 | 0·026 |
| 93952 | 2672·45 | Acetyl-ATKAVCVLKGDGPVQGIINFEQKES | Superoxide dismutase 1 | 2 | 26 | 1508·29 | 867·98 | 1·74 | 0·018 |
| 99116 | | | | | | 3·89 | 90·78 | 0·04 | 0·004 |
| 99658 | 2900·32 | GGSLPSEEKGHPQEESEESNVSMASLGE | Secretogranin-1 (Chromogranin B) | 296 | 323 | 83·35 | 8·52 | 9·78 | 0·021 |
| 100268 | | | | | | 298·79 | 2772·95 | 0·11 | 0·00009 |
| 103061 | 3079·61 | QASQAPPVPGPPPGAPHTPQTPGQGGPASTPAQ | NR2F2 Nuclear Receptor Subfamily 2 Group F Member 2 | | | 617·15 | 2517·87 | 0·25 | 0·002 |
| 103464 | | | | | | 0·71 | 77·81 | 0·01 | 0·001 |
| 105050 | | | | | | 274·50 | 1001·94 | 0·27 | 0·023 |
| 107641 | | | | | | 99·95 | 499·75 | 0·20 | 0·00009 |
| 113028 | | | | | | 1026·16 | 1728·45 | 0·59 | 0·042 |
| 114666 | | | | | | 163·64 | 33·12 | 4·94 | 0·023 |
| 118785 | 3677·743 | GPPGAIGPKGDRGFPGPLGEAGEKGERGPPGPAGSRGLP | Collagen alpha-1(VII) chain | | | 749·87 | 1882·40 | 0·40 | 0·021 |
| 124463 | 3880·04 | PPGRPEAQPPPLSSEHKEPVAGDAVPGPKDGSAPEVRGA | Neurosecretory protein VGF | 24 | 62 | 6129·44 | 3381·88 | 1·81 | 0·002 |
| 131863 | | | | | | 217·78 | 85·80 | 2·54 | 0·009 |
| 167142 | | | | | | 4133·79 | 1154·01 | 3·58 | 0·011 |
| 167697 | | | | | | 480·14 | 89·84 | 5·34 | <0·001 |
| 167938 | | | | | | 2427·14 | 451·22 | 5·38 | 0·004 |
| 168422 | | | | | | 159·22 | 12·62 | 12·61 | 0·001 |
| 168945 | | | | | | 3194·57 | 242·90 | 13·15 | 0·001 |

It is important to understand that the identification of peptides in our study was performed without previous protease digestion in a preparatory one-pot approach and holistically (capturing all peptides up to 20 kDa molecular mass) [13]. This approach distinguishes the present

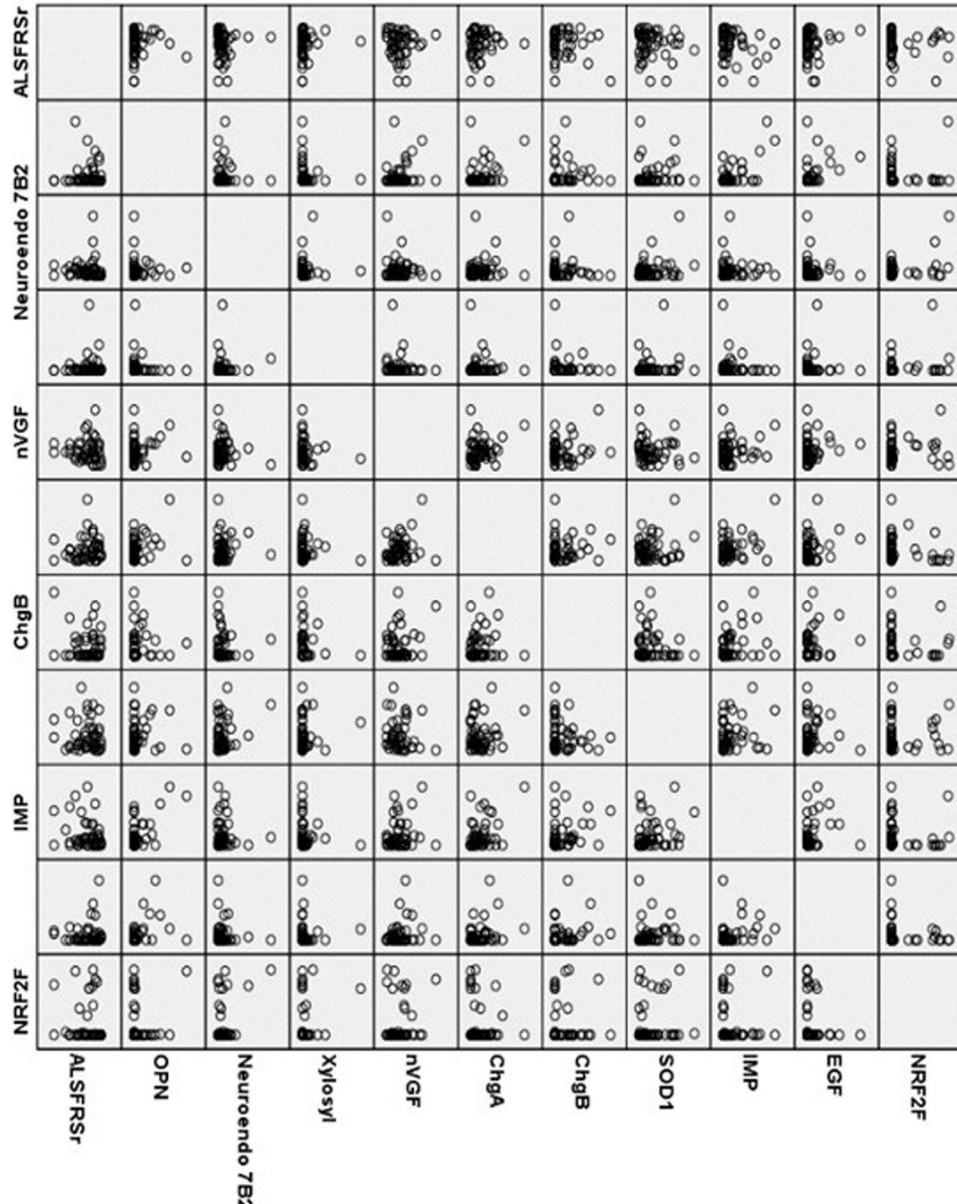

**Fig 1. Matrix associations between sequences peptides and ALS severity scale.** For multiple nVGF and OPN peptides, no association differences were detected and only one peptide is displayed.

study from most previous studies on proteomics in ALS CSF. The peptide sequences identified by this novel approach can provide further insights into disease pathophysiology and may possibly serve as future biomarkers after further investigation of underlying parenteral proteins by assay methods for whole proteins (mainly ELISA).

The identified peptides in our dataset are only partially new compared to those in previous studies [16,17]. The differences may be due to enhanced sensitivity of the utilized peptide quantification methods or the fact that the biomarker discovery method used in the majority of earlier studies is different from our holistic and non-digestive peptidomics approach.

In Table 3, the potential pathophysiological background of the identified biomarker peptides and comparison with the literature are depicted.

Changes in expression of the neurosecretory nerve growth factor inducible protein VGF deregulation have been described in the context of ALS, although with some inconsistencies [22].

We found two out of three peptides upregulated and in much higher abundance (absolute quantity making up more than 99% of all peptide mass) arguing for a resulting upregulation of total VGF in our dataset.

VGF is a secreted polypeptide and cleavage by prohormone convertases 1 and 2 generates a number of peptides with neuronal tissue-specific localization and diverse effects including modulation of neurogenesis, energy expenditure and reduced expression in spinal cord tissue and CSF of ALS patients [35,36]. VGF-derived peptides had neuroprotective effects in the mutant SOD1 G93A ALS mouse model and in the motor neuron-derived NSC-34 cell line expressing SOD1$^{G93A}$ serving as an ALS in vitro model [21,37,38].

Upregulated VGF peptides in our dataset could therefore be indicative for degradation. Another explanation of contrary literature results could be justified in difficult analytical principles avoiding lytic steps in our methodology. Overall, we consider VGF degradation products over-concentrated in ALS CSF due to increased degradation of the parental protein and subsequent lack of neuroprotective effect. Because the quantitative proteomic effect is huge, our data back the hypothesis of lacking intact VGF being a major promoting factor in the pathogenesis of ALS as determined by in vitro [39] and in vivo experiments [21].

**Table 3. Peptides with parental proteins, fold change vs. non-ALS controls, and differential regulation data from the literature.** Take notice of opposite regulation of different VGF and OPN peptides.

| This dataset | | Literature | |
|---|---|---|---|
| | **fold change** | **Regulation** | |
| **Integral membrane protein 2B** | 1·49 | | **Genetic variations [18]** |
| **Neurosecretory protein VGF** | 0·21 | ↑ | **4·8 kDa protein higher in pediatric patients with encephalopathy and seizures, (Seldi TOF-MS) [19,20], neuroprotective effects in ALS mouse model [21], described as biomarker in neurodegenerative diseases [22,23]** |
| **Neurosecretory protein VGF** | 1·87 | | |
| **Neurosecretory protein VGF** | 1·81 | | |
| **Osteopontin OPN** | 0·27 | ↑ | **Higher expression in ALS motoneurons [24], motoneurons have differential OPN vulnerability [25], levels higher in ALS sera [26]** |
| **Osteopontin OPN** | 5·77 | | |
| **Neuroendocrine protein 7B2 (Secretogranin-V)** | 4·01 | ↑ | **Increased (Seldi TOF-MS) [16]** |
| **EGF-containing fibulin-like extracellular matrix protein 1** | 5·15 | ↑ | **EFEMP 1 enhanced in brain lesions of amyloidosis pts [27], increased in serum of ALS pts [28]** |
| **Xylosyltransferase 1 XT-1** | 0·48 | | **None** |
| **Chromogranin-A Chg-A** | 0·54 | ↓ | **Modestly decreased, not discriminatory [29], produced by Golgi apparatus and reduction corresponds to the fragmentation of the Golgi complex [30], Cg-A may act chaperone-like and promote secretion of toxic SOD1 mutants, decreased through disease activity [31]** |
| **Superoxide dismutase SOD-1** | 1·74 | ⇅ | **Indifferent concentrations found by ELISA [32] and activity-labeled proteomics [28,32,33]** |
| **Secretogranin-1 (Chromogranin B) Chg-B** | 9·78 | ↓ | **Chromogranin B P413L variant as risk factor and modifier of disease onset and ameliorated staining of core particles in motoneurons [34]** |
| **NR2F2 Nuclear Receptor Subfamily 2 Group F Member 2** | 0·25 | | **None** |
| **Collagen alpha-1(VII) chain** | 0·40 | | **None** |

Mutations in membrane protein 2B (ITM2B) have been identified as cause of dementias and complex neurodegenerative phenotypes with no reports on the proteome so far [18,40]. We found the corresponding gene product only modestly up-regulated.

Several manuscripts report on association of the proinflammatory cytokine osteopontin / secreted phosphoprotein 1 (SSP-1) with ALS [24]. Reduced neuronal expression in the sensorimotor cortex and spinal cord of ALS patients has been reported. Moreover, it has also been found differentially expressed in microglia in an ALS rodent model and been suggested to contribute to the late stage of neurodegeneration by OPN induced MMP-9 up-regulation, CD44-mediated astrocyte migration and microglial phagocytosis [25,41]. Upregulation in the CSF has been reported by several proteomic studies before [26,36]. We confirmed this finding by proofing a massively upregulated peptide fragment 87061 in parallel with a low-abundant fragment 16933, which seems to be of regulative nature.

Superoxide dismutase (SOD) 1 was the first ALS gene identified in 1993 with mutations resulting in, and increased degradation and misfolding [42]. Increase in SOD derived peptides and consequently reduced SOD activity may be a key feature of familial mutant SOD1-associated ALS, exerted via mitochondrial dysfunction and oxidative stress [43] but alterations of SOD1-dependent pathways and SOD1 misfolding have also been described in sporadic ALS [44,45].

Hypothetically, SOD degradation might be influenced by the extended complexation with secretogranins (Chromogranin A and B), which we found differentially regulated in our datasets. Co-localization and interaction of CHGC and B with mutant SOD1 was seen in ALS mice [31], and interestingly, also interaction of oxidized wild type SOD1 with CHGB and subsequent motor neuron death has been observed [46,47] A variant in Chromogranin B has been identified as disease modifier of mutant SOD1-ALS and in one study also of sporadic ALS in a Canadian cohort [34,48]. The substantial (almost 10fold) upregulation of a Chromogranin B peptide, but not A (which was found down-regulated) following our non-differentiating approach seems to be the major pathophysiological nod in terms of disease activity, possibly stimulated by a de-differentiated SOD protein [43]. We found one SOD peptide in ALS CSF, which was only modestly upregulated. This medium-changed regulative pattern in our dataset indicates, that functional protein capabilities might not been accompanied by quantitative expression measures. Previous studies in ALS CFS mainly describe downregulation of chromogranin A but without sufficient discriminatory power [29,36] and reduction has also been detected in CSF of patients with Alzheimer's disease [49].

We also detected an upregulation of a peptide sequence of another granin, Neuroendocrine protein 7B2 (Secretogranin-V) which correlated with a previous SELDi-TOF proteomic study [16].

EGF-containing fibulin-like extracellular matrix protein 1 (EFEMP 1) also was markedly upregulated in our patient cohort, in line with a previous study showing an increase in ALS patient serum [28]. While no experimental data on a pathophysiological role in ALS exist so far, it may be of interest that it has been identified as a component of venous amyloid deposits in amyloidosis [27].

The remaining peptide sequences altered i.e. downregulated in the ALS patient cohort (Xylosyltransferase 1 XT-1, NR2F2 Nuclear Receptor Subfamily 2 Group F Member 2, Collagen alpha-1(VII) chain) have so far not yet been described in the context of ALS, neither as disease markers in patient material nor in cellular or animal models.

A limitation of our investigation is the lack of thorough analysis of the expressions of whole parental proteins in ALS versus control CSF to fully clarify the relevance of the observed up- and downregulated factors. Furthermore, the study lacked validation cohort that enables independent validation of the identified peptide biomarkers in ALS patients to confirm the findings.

However, after total crass-validation, we received as sensitivity of 96% and a specificity of 94%, with an accuracy of 95%. Further limitations lie in missing genetic nformation in 76% of cases.

Based on our findings, the observed alterations in ALS CSF proteome must now be confirmed by ELISA analysis. They then should be further explored in studies in larger ALS patient and control cohorts to evaluate potential correlations with disease phenotypes and their suitability as diagnostic and prognostic markers.

Regarding the pathophysiological implications summarized in Table 3, they should also trigger future therapeutic studies in vitro und in vivo ALS models aiming to restore physiological levels of the respective proteins by gene therapeutic or pharmacological measures and the potential beneficial impact on disease pathology.

In sake of rapid progress of further research on ALS biomarkers and identification of novel therapeutic targets, however, we aimed to stimulate protein identifications by rapid publication of our sequences identified by a novel holistic approach.

## Acknowledgments

This research was stimulated by the suffering of patients from the devastating disease. We are deeply grateful to Jochen S. being one of them who encouraged us to contribute to hopefully future treatments. Jochen passed away on January 14th, 2024.

## Author Contributions

**Conceptualization:** Susanne Petri, Harald Mischak, Joachim Beige.

**Data curation:** Justyna Siwy, Harald Mischak, Joachim Beige.

**Formal analysis:** Justyna Siwy, Agnieszka Latosinska, Julia Raad, Petra Zürbig, Harald Mischak, Joachim Beige.

**Investigation:** Susanne Petri, Justyna Siwy, Thomas Skripuletz, Harald Mischak, Joachim Beige.

**Methodology:** Justyna Siwy, Agnieszka Latosinska, Julia Raad, Petra Zürbig, Harald Mischak.

**Project administration:** Harald Mischak, Joachim Beige.

**Resources:** Thomas Skripuletz, Harald Mischak.

**Software:** Harald Mischak.

**Supervision:** Harald Mischak, Joachim Beige.

**Validation:** Joachim Beige.

**Visualization:** Joachim Beige.

**Writing – original draft:** Rea Lumi, Susanne Petri, Harald Mischak, Joachim Beige.

**Writing – review & editing:** Rea Lumi, Susanne Petri, Justyna Siwy, Agnieszka Latosinska, Julia Raad, Petra Zürbig, Thomas Skripuletz, Harald Mischak.

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
