## [Decision Letter · Decision Letter 0]

26 Feb 2024

PONE-D-24-00162Small peptide CSF fingerprint of Amyotrophic Lateral SclerosisPLOS ONE

Dear Dr. Beige,

Thank you for submitting your manuscript to PLOS ONE. After careful consideration, we feel that it has merit but does not fully meet PLOS ONE’s publication criteria as it currently stands. Therefore, we invite you to submit a revised version of the manuscript that addresses the points raised during the review process.

Based on the reviewers' suggestions, the paper needs minor revision.  The reviewers' comments can be found below.

We look forward to receiving your revised manuscript.

Kind regards,

Tanja Grubić Kezele, Ph.D., M.D.

Academic Editor

PLOS ONE

“S. Petri received honoraria as speaker/consultant from Biogen GmbH, Roche, Novartis, Teva, Cytokinetics Inc., Desitin, Italfarmaco, Ferrer, Amylyx, and Zambon; and grants from DGM e.V, Federal Ministry of Education and Research, German Israeli Foundation for Scientific Research and Development, EU Joint Program for Neurodegenerative Disease Research. J. Beige received funding from GSK and German Federal Ministries of Research and Health.”

Reviewers' comments:

Reviewer's Responses to Questions

**Comments to the Author**

1. Is the manuscript technically sound, and do the data support the conclusions?

Reviewer #1: Yes

Reviewer #2: Partly

2. Has the statistical analysis been performed appropriately and rigorously? 

Reviewer #1: Yes

Reviewer #2: No

3. Have the authors made all data underlying the findings in their manuscript fully available?

Reviewer #1: Yes

Reviewer #2: No

4. Is the manuscript presented in an intelligible fashion and written in standard English?

Reviewer #1: Yes

Reviewer #2: Yes

5. Review Comments to the Author

Reviewer #1: ALS is an important neurodegenerative disease, which is also caused by neuron damage caused by abnormal protein aggregation, and the pathological factors are complex. Protein components in CSF may contain molecular information and diagnostic markers useful for guide therapy. The authors used specific changes of low molecular weight proteins or peptides in cerebrospinal fluid to search for evidence of decoding ALS molecular pathology to guide intervention. Although some different proteins were found, there was no obvious association with disease progression, but it also provided a new research method for ALS research. The concept of the study is interesting, only preliminary studies were performed. So I recommend it to be published in PLOS ONE after minor revisions.

Reviewer #2: The focus of the study seems to investigate whether specific changes in cerebrospinal fluid (CSF) peptides or low molecular weight proteins could serve as biomarkers to ALS and potentially guide therapeutic interventions.

Major comments:

-In the Wilcoxon rank-sum test, the authors say they have adjusted for multi testing, but they have considered peptides with P < 0.05. Is it adjusted P or nominal P-value? Please be specific on which P-value was considered for statistical significance.

-While the authors may have used to the best of their samples size, they surely lacked validation cohort that enables independent validation of the identified peptide biomarkers in ALS patients to confirm these findings and increase confidence in their approach.

-Further analysis needed but not required for this publication is longitudinal assessment profiles over time could provide valuable information on disease progression and the dynamic changes in peptides biomarkers eliminating the “by chance nature of these findings” as some of the peptides found show very low fold change values.

-While the authors mentioned 76% ALS samples lacked genetic information, the validity of these findings still reflect underlying biological processes and pathophysiological mechanisms of ALS. However, my recommendation would be for the authors to follow up this paper with a much thorough investigation incorporating genetic information in their next analysis.

Minor comments:

-Please provide a link to GitHub code in this paper for all the analysis done for reproducible and responsible research.

-Also provide data repository links or accession numbers to spectral data, metadata (with PII removed) and additional information of Statistical methods/softwares.

-Provide additional information on the peptide sequencing methods used including details on the accuracy and reliability of sequencing results.

-Provide future research directions from these findings, including potential avenues for further investigation and clinical translation of identified biomarkers.

6. PLOS authors have the option to publish the peer review history of their article (what does this mean?). If published, this will include your full peer review and any attached files.

Reviewer #1: No

Reviewer #2: No

---

## [Author Response · Author response to Decision Letter 0]

19 Mar 2024

Manuscript revision (PONE-D-24-00162)

Title: Small peptide CSF fingerprint of Amyotrophic Lateral Sclerosis

Dear Editor,

the decision letter for our manuscript entitled “Small peptide CSF fingerprint of Amyotrophic Lateral Sclerosis”; PONE-D-24-00162) has been received and very much appreciated.

We would like to thank the reviewer for the positive, valuable feedback. We have studied the comments carefully and have made corresponding modifications that we hope will meet with your approval. 

Revised parts are visible in yellowmark.

Please find our responses to the reviewer’s comments in a point-by-point manner below.

By resubmission, we confirm that all author details on the revised version are correct, that all authors have agreed to authorship and order of authorship and that all authors have the appropriate permissions and rights to the reported data.

We believe the modifications have resulted in an improved manuscript and hope that the revised version is now suitable for publication in PLOS ONE.

Yours sincerely, 

Responses to reviewer’s comments

Reviewer #1: ALS is an important neurodegenerative disease, which is also caused by neuron damage caused by abnormal protein aggregation, and the pathological factors are complex. Protein components in CSF may contain molecular information and diagnostic markers useful for guide therapy. The authors used specific changes of low molecular weight proteins or peptides in cerebrospinal fluid to search for evidence of decoding ALS molecular pathology to guide intervention. Although some different proteins were found, there was no obvious association with disease progression, but it also provided a new research method for ALS research. The concept of the study is interesting, only preliminary studies were performed. So I recommend it to be published in PLOS ONE after minor revisions.

Reviewer #2: The focus of the study seems to investigate whether specific changes in cerebrospinal fluid (CSF) peptides or low molecular weight proteins could serve as biomarkers to ALS and potentially guide therapeutic interventions.

Major comments:

-In the Wilcoxon rank-sum test, the authors say they have adjusted for multi testing, but they have considered peptides with P < 0.05. Is it adjusted P or nominal P-value? Please be specific on which P-value was considered for statistical significance.

Answer: As mentioned in the manuscript (“Using the Wilcoxon rank-sum test followed by adjustment for multiple testing with the false-discovery rate method presented by Benjamini and Hochberg [14], adjusted P-values were calculated based on the comparison between cases and healthy controls.”), we used the adjusted p-values.

-While the authors may have used to the best of their samples size, they surely lacked validation cohort that enables independent validation of the identified peptide biomarkers in ALS patients to confirm these findings and increase confidence in their approach.

Answer: We agree and mentioned this in the discussion section as a shortcoming of the study: “Furthermore, the study lacked validation cohort that enables independent validation of the identified peptide biomarkers in ALS patients to confirm the findings. However, after total cross-validation, we received as sensitivity of 96% and a specificity of 94%, with an accuracy of 95%.”

-Further analysis needed but not required for this publication is longitudinal assessment profiles over time could provide valuable information on disease progression and the dynamic changes in peptides biomarkers eliminating the “by chance nature of these findings” as some of the peptides found show very low fold change values.

-While the authors mentioned 76% ALS samples lacked genetic information, the validity of these findings still reflect underlying biological processes and pathophysiological mechanisms of ALS. However, my recommendation would be for the authors to follow up this paper with a much thorough investigation incorporating genetic information in their next analysis.

Answer: We thank the reviewer for this important suggestion. Genetic testing has now been implemented into the diagnostic workup of each newly diagnosed patient and all patients still followed up at our department are also offered genetic testing so that we will be able to incorporate this information into future analyses.

Minor comments:

-Please provide a link to GitHub code in this paper for all the analysis done for reproducible and responsible research.

Answer: The used methods (e.g. Wilcoxon test) are standard methods, which are integrated in commercial software (e.g. MedCalc). The other analyses we made by use of mass spectrometric software tools, which are also commercial available. Therefore, we can not provide a link to GitHub code.

-Also provide data repository links or accession numbers to spectral data, metadata (with PII removed) and additional information of Statistical methods/softwares.

Answer: We have uploaded the CE-MS data on a repository (DOI: 10.5281/zenodo.10716167). Furthermore, we included the accession numbers in the table 2. The clinical data cannot be uploaded due to uncertainty regarding GDPR issues. Information of statistical software (R) is now included in the Method section.

-Provide additional information on the peptide sequencing methods used including details on the accuracy and reliability of sequencing results.

Answer: Unfortunately, we do not know which information is still missing. The accuracy and reliability of the sequencing results are also already given in the Method section (Sequencing of peptides): “FDR was set to 1%, precursor mass tolerance was 5 ppm and fragment mass tolerance 0,05 Da. For further validation of obtained peptide sequences, the correlation between peptide charge at the working pH of 2 and CE-migration time was utilised to minimise false-positive derivation rates. Calculated CE-migration time of the sequence candidate based on its peptide sequence (number of basic amino acids) was compared to the experimental migration time.”

-Provide future research directions from these findings, including potential avenues for further investigation and clinical translation of identified biomarkers.

Answer: While the pathophysiological implications of our data and previous results from the literature have been discussed in the discussion section and summarized in table 3, we have now added further research directions that arise from our findings of differentially expressed peptides in ALS CSF in the last paragraph of the discussion.

---

## [Decision Letter · Decision Letter 1]

2 Apr 2024

Small peptide CSF fingerprint of Amyotrophic Lateral Sclerosis

PONE-D-24-00162R1

Dear Dr. Beige,

We’re pleased to inform you that your manuscript has been judged scientifically suitable for publication and will be formally accepted for publication once it meets all outstanding technical requirements.

Kind regards,

Weidong Le

Academic Editor

PLOS ONE

Additional Editor Comments (optional):

Reviewers' comments:

Reviewer's Responses to Questions

**Comments to the Author**

1. If the authors have adequately addressed your comments raised in a previous round of review and you feel that this manuscript is now acceptable for publication, you may indicate that here to bypass the “Comments to the Author” section, enter your conflict of interest statement in the “Confidential to Editor” section, and submit your "Accept" recommendation.

Reviewer #1: All comments have been addressed

Reviewer #2: (No Response)

2. Is the manuscript technically sound, and do the data support the conclusions?

Reviewer #1: Yes

Reviewer #2: (No Response)

3. Has the statistical analysis been performed appropriately and rigorously? 

Reviewer #1: Yes

Reviewer #2: (No Response)

4. Have the authors made all data underlying the findings in their manuscript fully available?

Reviewer #1: Yes

Reviewer #2: (No Response)

5. Is the manuscript presented in an intelligible fashion and written in standard English?

Reviewer #1: Yes

Reviewer #2: (No Response)

6. Review Comments to the Author

Reviewer #1: I have reviewed this revised manuscript. The authors have well improved the manuscript and have properly addressed my concerns. So, this work can be recommended now to be accepted for publication in PLOS ONE.

Reviewer #2: (No Response)

7. PLOS authors have the option to publish the peer review history of their article (what does this mean?). If published, this will include your full peer review and any attached files.

Reviewer #1: No

Reviewer #2: No
